# Study of the Antimicrobial Activity of the Human Peptide SQQ30 against Pathogenic Bacteria

**DOI:** 10.3390/antibiotics13020145

**Published:** 2024-02-01

**Authors:** Michela Di Napoli, Giusy Castagliuolo, Sara Pio, Ilaria Di Nardo, Teresa Russo, Dario Antonini, Eugenio Notomista, Mario Varcamonti, Anna Zanfardino

**Affiliations:** 1Department of Biology, University of Naples Federico II, 80126 Naples, Italy; michela.dinapoli@unina.it (M.D.N.); giusy.castagliuolo@unina.it (G.C.); sa.pio@studenti.unina.it (S.P.); ilaria.dinardo@unina.it (I.D.N.); dario.antonini@unina.it (D.A.); notomist@unina.it (E.N.); varcamon@unina.it (M.V.); 2IPCB—Institute for Polymers, Composites and Biomaterials, National Research Council of Italy, 80125 Naples, Italy; teresa.russo@cnr.it

**Keywords:** antimicrobial activity, human peptide, biological properties

## Abstract

Given the continuous increase in antibiotic resistance, research has been driven towards the isolation of new antimicrobial molecules. Short, charged, and very hydrophobic antimicrobial peptides have a direct action against biological membranes, which are less prone to developing resistance. Using a bioinformatic tool, we chose the SQQ30 peptide, isolated from the human SOGA1 protein. The antimicrobial activity of this peptide against various Gram-negative and Gram-positive bacterial strains and against a fungal strain was studied. A mechanism of action directed against biological membranes was outlined. When administered in combination with the antibiotic ciprofloxacin and with the TRS21 (buforin II), another antimicrobial peptide, SQQ30 can be used with a lower MIC, showing additivity and synergism, respectively. Particularly interesting is the ability of SQQ30 to bind LPS in Gram-negative strains, preventing the eukaryotic cell from releasing inflammatory mediators. Our study indicates SQQ30 as a novel and promising antimicrobial agent.

## 1. Introduction

Before the appearance of antibiotics, humans were defenseless against many types of infections, which very often led to the death of millions of people. The advent of antibiotic therapy has revolutionized the world, saving millions of lives. Today, less than a century after the discovery of antibiotics, we face a major global health problem: antibiotic resistance (AMR). The World Health Organization has identified AMR as one of the most important public health threats of the 21st century and a major global health challenge to be addressed [1,2]. This is not only due to spontaneous mutations in chromosomal genes selected via drug pressure but can also be the result of a horizontal gene transfer, the movement of genetic information between organisms via transformation, conjugation, or transduction [3]. To prevent drug action and microorganism death, bacteria have developed various resistance mechanisms. Based on the involved biochemical pathway, we can observe (i) the modification or degradation of the antimicrobial molecules, (ii) the preclusion of reaching and the protection of the target sites (decreasing the penetration and active extrusion of the antimicrobial), the (iii) modification and/or bypass of target sites [4].

Although antimicrobial resistance is a natural phenomenon, the process has been greatly enhanced due to a variety of contributing causes: self-medication (some antibiotics can be bought without a prescription and are improperly dosed); the use of antibiotics in animal feed and agriculture to promote the growth of animals and food; inappropriate antibiotic prescriptions by healthcare providers; and the use of antibiotics with a broader spectrum than needed [5,6]. The antibiotic effectiveness decline has caused an increase in mortality and morbidity, leading to catastrophic consequences not only for individuals but also for the healthcare system and society as a whole [7]. The remedies that can be used to manage this problem include antimicrobial peptides (AMPs), also known as host defense peptides (HDPs). They are small polypeptide molecules, typically consisting of about 10–50 amino acids, which show a net positive charge—ranging from +2 to +11—and contain a large number of hydrophobic residues (typically 50%) [8,9]. Most have a net positive charge at a neutral pH that allows them to interact, through electrostatic interactions, with the cytoplasmic membranes negatively charged of Gram (−) and Gram (+) bacteria. Furthermore, some AMPs are active against fungi, viruses, or other parasites [10]. After membrane binding, AMPs alter integrity through pore formation. This occurs through different patterns: “barrel-stave”, “carpet”, “toroidal pore”, and “aggregate”, which result in the leakage of cell contents and their death [11,12]. This interaction does not require specific targets, so it is rarer for bacteria to develop resistance to AMPs. Some AMPs are able to cross the membrane without damaging it, but they attack cytoplasmic targets and influence cellular processes; they inhibit transcription, translation, and protein synthesis, they alter cell wall formation, the synthesis of nucleic acids, and cell division, and they block proteases’ activity [13,14].

Although AMPs perform direct antimicrobial, antiviral, and/or antifungal activity, they also perform other biological activities, such as immunomodulation, the inhibition of lipopolysaccharide (LPS)-induced inflammation, the improvement of wound healing, and antitumor activity [15]. This combination of characteristics, in addition to little or no development of resistance against microorganisms and their lack of toxicity, makes AMPs promising molecules to limit the use of conventional antibiotics, thus coping with the antibiotic resistance problem. In addition, studies on the synergism between AMPs and conventional antibiotics are receiving a positive response.

Synergism aims to decrease the effective concentration of the active molecules, reduce possible toxic side effects, and limit the spreading of resistance, which is very often associated with monotherapy [16]. In this study, we characterized the biological properties of the SQQ30 peptide: antimicrobial, LPS binding, immunomodulatory activity, and its role in oxidative stress. We outline the peptide’s mechanism of action, its ability to lower the MIC of antibiotics, and its ability to act in combination with other peptides.

## 2. Results

### 2.1. Identification of the Cryptic Antimicrobial Region in SOGA1

The cryptic peptide SQQ30 was selected using a previously developed in silico tool designed to identify cryptic AMP-like regions hidden in the sequence of proteins [15]. This tool is based on the finding that the antimicrobial potency of AMPs (defined as Log(1000/MIC)) is linearly correlated to an antimicrobial score (A.S.), depending on three peptide-related variables, namely net charge, hydrophobicity, and length, and on four strain-dependent variables determined via titration. Using parameters determined for *S. aureus* and *P. aeruginosa* [15], a linear correlation was observed for scores from about 6 to 12. For scores higher than 12, the linear relationship may no longer be valid, and an increase in the score values does not necessarily indicate higher antimicrobial activity. By calculating the A.S. values of all the peptides in a protein through a sliding-window approach, it is possible to identify the presence and boundaries of cryptic AMPs. This approach, initially applied to about 3000 human-secreted proteins, allowed the identification of many new cryptic human AMPs, including AMPs from apolipoproteins E [17,18] and B [19] and human pepsinogen A3 [20]. Later on, the analysis was extended to the whole human proteome, allowing the identification of several hundreds of further cryptic AMPs [21]. SOGA1 is an unusual protein included in the initial pool of analyzed human-secreted sequences. It is a 1423-residue-long intracellular protein containing a secreted 80 kDa C-terminal domain (probably starting from residue 689) [22]. It has been hypothesized that the secretion of the C-terminal domain is mediated by a predicted internal signal peptide spanning from positions 667 to 686 [22]. Very interestingly, the sliding-window analysis of this region of SOGA1 (Appendix A) showed a local maximum (A.S. = 14.2) that corresponded exactly to the predicted internal signal peptide (the left arrow) and an absolute maximum (the two arrows on the right) corresponding to peptides 667–691 (A.S. = 15.9) and 667–692 (A.S. = 15.8). All the local maxima showed A.S. values above the linear correlation range; therefore, it can be expected that the corresponding peptides should possess similar antimicrobial activity. However, we noticed that regions 665–687, including the predicted signal peptide and three flanking glutamine residues, can fold as a long amphipathic helix, as shown in the helical wheel projection (Appendix A) and the structural model (Appendix A). Because amphipathicity is a well-known key determinant of AMPs’ ability to interact with and damage membranes, we decided to test the antimicrobial properties of regions 664–693, containing the peptide with the highest A.S. value, and residues Gln665 and Gln666, which contribute to forming the above-mentioned amphipathic helix (Appendix A). The final selected peptide also includes two flanking serine residues, Ser664 and Ser693. Ser664 was included to avoid glutamine at position 1, as N-terminal glutamine residues frequently undergo cyclization to pyroglutamate, thus causing heterogeneity. Ser693 was included to increase the flexibility of the C-terminal carboxylate.

### 2.2. Antimicrobial Activity of SQQ30

To study the antimicrobial features of the SQQ30 peptide on the selected bacterial strains, a cell survival assay was performed. Specifically, the Gram (−) *Escherichia coli*, *Pseudomonas aeruginosa* and *Salmonella* Typhimurium bacteria and the Gram (+) *Staphylococcus aureus*, *Streptococcus oralis*, *Streptococcus mutans*, *Staphylococcus pasteuri*, and *Staphylococcus warneri* bacteria were incubated with different concentrations of the peptide (1, 5, 10, and 20 µM), and dose-response curves were obtained.

Figure 1 shows that the peptide performs significant antimicrobial activity; in fact, there was a considerable reduction in bacterial survival (expressed as a percentage on the y-axis) even at low concentrations (1 µM), the activity was stronger towards *E. coli*, *P. aeruginosa*, *S. aureus*, *S. warneri*, and *S. pasteuri*. Bacterial mortality increased in a dose-dependent manner.

The SQQ30 minimum inhibitory concentration (MIC) for bacterial growth was calculated using the microdilution method. Table 1 shows the MIC values obtained. In agreement with the previous experiments, the lowest MIC values (1.5–7 µM) were obtained against *E. coli*, *P. aeruginosa*, *S. aureus*, *S. warneri*, and *S. pasteuri*. This result confirmed that the antimicrobial action of SQQ30 was directed against both Gram (−) and Gram (+) bacteria. To complete the analysis of the SQQ30 antimicrobial properties, the antifungal activity was evaluated via a viable count assay. From the dose-response curves reported in Appendix A, it is possible to observe that the peptide showed good antifungal activity.

The microdilution method to obtain the SQQ30 MIC value against *C. albicans* was calculated, as reported in Appendix A; this value was 40 µM.

### 2.3. FIC Index Determination between SQQ30, Ciprofloxacin, and TRS21

Given the antibiotic resistance concern and the possibility that using fewer antibiotics might slow down the onset of resistance, we wanted to test whether the antimicrobial activity of the SQQ30 peptide could have synergistic effects in combination with an antibiotic and/or the other peptide, using conventional antibiotics at the lowest possible concentrations or even avoiding them.

The FIC index value was calculated as a predictor of synergy using SQQ30 and TRS21, known also as buforin II (isolated from Amphibian Histone protein H2A) peptides and the conventional antibiotic ciprofloxacin combined against *E. coli*, according to the chessboard scheme. Before proceeding with this experiment, we calculated the TRS21 and ciprofloxacin MIC values against *E. coli*, which respectively resulted in 100 and 1.66 µM. Table 2 indicates the MIC values of the two peptides and antibiotics alone or in combination with each other. This experiment showed a synergistic interaction between SQQ30 and TRS21 (FIC index = 0.45) and an additive interaction between SQQ30 and ciprofloxacin (FIC index = 0.95). The peptides and antibiotic concentration to be used to inhibit microbial growth were significantly decreased.

### 2.4. Study of SQQ30′s Mechanism of Action Using a Microscopy Approach

Since the literature states that AMPs mainly exert their action in damaging bacterial membranes, we studied SQQ30′s possible mechanism of action underlying its antimicrobial activity via microscopy techniques.

To search for the possible bacterial target, we carried out fluorescence microscopy (FM) studies, using 4′,6-diamidino-2-phenylindole dihydrochloride (DAPI) and propidium iodide (PI), on the E. coli indicator strain.

Specifically, DAPI freely crossed the membrane, bound to DNA, and stained the cells blue. PI, on the other hand, was unable to pass through intact bacterial membranes, thus red-staining only the cells with damaged membranes. 

In particular, after 4 h of the SQQ30 treatment (1 µM), a fluorescence microscope observation allowed us to highlight that the peptide caused damage to the cell membrane (Figure 2). In fact, cells treated with SQQ30 (Figure 2C,D), compared to the control—consisting of a cell culture alone (Figure 2A,B)—showed a marked red fluorescence also associated with bacterial aggregation, which was not observed in the untreated sample. The bacterial aggregation was probably due to the nature of the peptide itself. In fact, SQQ30 is a highly hydrophobic peptide with positive charges. This analysis suggests that the antimicrobial action of SQQ30 against *E. coli* could probably be exerted through membrane damage.

To deepen the data obtained via the fluorescence microscopy evaluations, we performed an analysis using scanning electron microscopy (SEM). To describe SQQ30′s antimicrobial activity via SEM, we replicated the experimental conditions of the previous study (concentration of 1 µM for 4 h against *E. coli*).

The SEM observations confirmed that the target of SQQ30 was the membranes (Figure 3). In fact, it is possible to observe that—when compared to untreated sample (Figure 3A)—treated cells (Figure 3B) appear crushed and emptied of cellular contents, losing their characteristic three-dimensionality.

This effect is related to the cellular emptying caused by damage at the membrane level. Therefore, we can say that this observation is in agreement with various studies in the literature and with our previous fluorescence microscopy analysis [23].

To complete the microscopy study, we performed transmission electron microscopy (TEM) observations. We maintained the same experimental conditions as the previous experiments.

As it is possible to observe in Figure 4, when compared to the control (untreated cells, Figure 4A), the treated cells (Figure 4B) appeared to take on an amorphous shape. In particular, the untreated cells appeared to be intact on their entire surface, whereas the cells treated with SQQ30 presented a structurally altered surface and seemed completely empty of their cellular content, showing the disintegration of the membrane. This confirms that there was damage at the membrane level.

### 2.5. Interaction between SQQ30 and Lipopolysaccharide

We performed an in vitro bacterial survival assay to evaluate the possible interaction between SQQ30 and *E. coli* LPS. The percentage of cell survival was calculated by counting the colonies present in the treated samples compared to the control (untreated). *E. coli* cultures were then incubated for 4 h with either LPS or SQQ30 or both. In Figure 5, it is possible to observe that the addition of *E. coli* LPS (0.5 µg/mL) did not influence bacterial survival (comparable to the control).

Instead, incubation with the peptide at a concentration of 1 μM led to approximately 85% cell mortality. By co-incubating SQQ30 with LPS, it is possible to measure a reduction in the cell mortality rate; the survival values are indeed comparable to those of the positive control. It is possible to conclude that an interaction between SQQ30 and LPS could be responsible for the antimicrobial peptide activity’s inhibition.

### 2.6. Study of the SQQ30 Immunomodulatory Activity

The ability of SQQ30 to scavenge from LPS pro-inflammatory activity was determined using murine macrophages (RAW 264.7 cells). When stimulated with LPS, RAW 264.7 cells release inflammation mediators, e.g., nitric oxide (NO) and several cytokines [24].

First, we tested the biocompatibility of SQQ30 on untreated RAW 264.7 cells via an MTT (3-(4,5-dimethylthiazol-2-yl)-2,5-diphenyltetrazolium bromide) assay. The peptide caused a limited decrease in cell viability (20–25%) only at concentrations higher than 10 µM (Appendix A).

Hence, the RAW 264.7 cells were treated with a fixed amount of LPS from *E. coli* (10 ng/mL) and increasing concentrations (0.1, 1, and 10 μM) of SQQ30 for 24 h. Colistin, a lipopeptide with a very strong binding to LPS [25], was used as a positive control. After the incubation, NO and tumor necrosis factor-α (TNF-α) were measured respectively via a Griess assay and ELISA. SQQ30 did not itself induce the release of NO and TNF-α in the absence of LPS (Figure 6), whereas it effectively inhibited the LPS-induced release of the two inflammation mediators (Figure 6).

It is worth noting that SQQ30 completely re-established basal levels of pro-inflammatory mediators at 10 μM, a concentration with no toxic effect on cells. Therefore, our results indicate that SQQ30 is an LPS scavenger with an efficacy comparable to that of colistin.

### 2.7. Anti-H_2_O_2_ Effect of SQQ30

For a future antimicrobial application, another interesting aspect is the ability of SQQ30 to inhibit viability and/or oxidative stress in eukaryotic cells. We evaluated the antioxidant activity of SQQ30 using a hydrogen-peroxide scavenging analysis. An increased percentage of H_2_O_2_ radical scavenging activity was observed upon SQQ30′s addition (Figure 7). The results shown in Figure 7 are expressed in Table 3 as IC_50_ values, that is, the concentration of the peptide conferring a 50% decrease in H_2_O_2_ radicals.

In detailed terms, therefore, the SQQ30 peptide in the concentration range between 1 and 20 μM eliminated the maximum OH radical by 65%; the anti-H_2_O_2_ activity achieved IC_50_ values equal to 17 μM.

### 2.8. SQQ30 Activity on Cell Viability and Oxidative Stress in Epithelial Cells

In order to assess SQQ30′s effect on cell viability and oxidative stress in eukaryotic epithelial cells, we performed an MTT assay with HaCat cells upon SQQ30 treatment. The obtained results show that the SQQ30 peptide caused a limited decrease in cell viability (10–15%) at the concentration of 20 µM for 48 h (Appendix A).

In addition, to assess SQQ30′s effect on oxidative stress, we pretreated HaCat cells with SQQ30 for 24 h. A reactive oxygen species (ROS) detection assay demonstrated that neither alone nor in concomitant treatment with TRS21 does SQQ30 induce oxidative stress (Figure 8). In addition, we tested whether SQQ30 may have an effect in the presence of hydrogen-peroxide-induced oxidative stress.

In addition, under induced oxidative stress conditions upon peroxidase hydrogen treatment, SQQ30 alone or with TRS21 does not increase ROS (Figure 8). These data demonstrate that SQQ30 moderately affects cell viability only at high concentrations, whereas it does not affect ROS under either basal or oxidative stress conditions in epithelial eukaryotic cells.

## 3. Discussion

In this study, we show the antimicrobial activity of SQQ30 peptide and its effect on epithelial eukaryotic cells. Our data highlight the potential application of this novel peptide in the fight against pathogenic bacteria. SQQ30 was selected from a pool of hundreds of potential antibacterial cryptic peptides, based on the calculation of an antimicrobial score (A.S.) that depended on three peptide-related variables: net charge, hydrophobicity, and length. As predicted, the peptide showed excellent antimicrobial properties against both Gram (−) and Gram (+) strains at low concentrations [26].

Our studies on the mechanism of action of SQQ30 were carried out using FM, SEM, and TEM. The results presented in this paper establish that the antimicrobial effect of SQQ30 depends on its ability to disrupt bacterial membranes. Hartmann et al. stated that microscopic studies are very important because they are needed as complementary techniques to obtain information on the action of AMPs [27]. They demonstrated the same result on *E. coli* and *S. aureus* after treatment with the β-stranded gramicidin S and the α-helical peptidyl-glycilleucine-carboxamide (PGLa) peptides, thus highlighting membrane damage [27]. The effect of SQQ30 on bacterial membranes was observed via microscopic imaging analysis; an indirect test performed on Gram-negative bacteria revealed the presence of an LPS–peptide relation, confirming a physical interaction between the peptide and the external membrane.

It is well known that most cationic AMPs have a significant ability to bind LPS. This is not surprising, given that they have an amphipathic structure and a high positive charge that mediate strong binding to anionic lipids. This is also the case for LPS.

Murin RAW 264.7 cells are commonly used to demonstrate the ability of antimicrobial peptides to act as LPS scavengers because they possess high-affinity receptors for these lipids, respond to their activation by secreting several anti-inflammatory molecules, and activate the immune system [28,29].

Most AMPs interact and destabilize both the external and internal membranes of bacteria [30,31], causing the formation of pores with a consequent rupture, loss of integrity, and increase in the permeability of the membrane, or they inhibit bacterial macromolecular functions [32]. Given the rapid ability of bacteria to mutate in order to respond to the action of antimicrobial agents, synergistic combinations appear to be truly promising strategies [33,34]. Indeed, synergistic combinations of antimicrobial molecules that have different targets may require two sets of independent and simultaneous mutations in bacteria [35]. These synergistic interactions can potentially increase the antimicrobial efficacy of an antibiotic and/or a peptide, counteract the phenomenon of antibiotic resistance, and reduce toxicity in the host since the concentrations of each antimicrobial component are lowered to obtain a broad antimicrobial effect [36].

Several studies have indicated that AMPs can act synergistically with each other or with conventional antibiotics, generating more efficient antibacterial activity. For example, antibiotic action against *C. difficile* is improved when both LL-37 and HBD3 are present [37,38]. In our study, the synergistic and additive properties of the SQQ30 peptide were demonstrated in combination with another peptide, TRS21, and with a conventional antibiotic, ciprofloxacin. TRS21 and ciprofloxacin were chosen because, unlike SQQ30, which targets membranes, they have an intracellular target (or intracellular targets). TRS21 is a peptide that translocates across bacterial membranes and interacts with nucleic acids; ciprofloxacin is a broad-spectrum fluoroquinolone that inhibits DNA gyrase and topoisomerase types II and IV, both located in the cytoplasm [39,40,41]. As reported in some studies [42], we hypothesize that the membrane destabilization and permeability produced via SQQ30 likely allow better access to TRS21 and ciprofloxacin. Additionally, intracellular ciprofloxacin and TRS21 can reduce the ability of bacteria to repair their membranes, and this may also help their synergy. Furthermore, in Gram-negative bacteria, there are efflux pumps that extrude antibiotics, consuming ATP. Damaging the bacterial membrane means altering the site of energy production (ATP) and reducing the functioning of the efflux pumps. Our results demonstrate that SQQ30 presents additive and synergistic properties in combination with both TRS21 and ciprofloxacin, improving their accessibility through the bacterial membrane.

Subsequently, attention was focused on the analysis of the possible activities to be correlated to this peptide, such as immunomodulatory activity. From the data obtained, it seems that SQQ30 is able to attenuate LPS-induced inflammation by reducing the level of NO and TNF-*α*. LPS is known to present in vivo complications related to an individual’s immune response. The release of LPS into the environment stimulates the activation of an organism’s immune cell response, leading to sepsis and septic shock [43].

In severe infections, a single antimicrobial agent may not sufficiently guarantee the patient’s recovery. The neutralization of LPS via AMPs is a crucial effect that occurs either through a non-specific electrostatic interaction with LPS or through an immunomodulatory event [29].

Moreover, intracellular ROS detection experiments have indicated that SQQ30 peptide does not affect oxidative stress either under basal or ROS-induced conditions in epithelial cells. However, our data do not show a significant antioxidant effect of SQQ30 alone or in combination with the TRS21 peptide. Many studies have demonstrated that antioxidant activities have been associated with an additive combination of AMPs with various properties and several abilities, such as removing free radicals, inactivating intracellular ROS, chelating transition metals, and suppressing the peroxidation of lipids [44].

This study provides novel insights regarding AMPs and their combinations for antimicrobial applications. SQQ30 shows an immunomodulatory activity that neutralizes LPS, which has negligible toxicity in eukaryotic cells.

All of our results provide evidence that SQQ30 could be a promising antimicrobial agent against bacterial infections in the future.

## 4. Materials and Methods

### 4.1. Antimicrobial Peptide (AMPs)

The SQQ30 peptide (Thermo Fisher Scientific, Monza, Italy) has a molecular weight of 3833.67 g/mol (SQQFKHNFLLLFMKLRWFLKRWRQGKVLPS), and the TRS21 peptide (Caslo ApS, Kongens Lyngby, Denmark) has a molecular weight of 2435.03 g/mol (TRSSRAGLQFPVGRVHRLLRK). The mass of the peptides was measured via mass spectroscopy (MS); it corresponds to the theoretical mass.

### 4.2. Bacterial Strains

The Gram (−) bacterial strains used were *Escherichia coli* DH5α, *Pseudomonas aeruginosa* PAO1 ATCC 15692, and *Salmonella* Typhimurium ATCC14028. The Gram (+) strains used were *Staphylococcus aureus* ATCC6538P, *Streptococcus oralis* CECT 8313, *Streptococcus mutans* ATCC 35668, *Staphylococcus pasteuri* (>99% identity with *S. pasteuri P4*) and *Staphylococcus warneri* (100% identity with *S. warneri MBSb5a*) isolated from the plaque of healthy patients. Furthermore, the fungus *Candida albicans* ATCC 14053 was used.

### 4.3. Antimicrobial Activity Assay

Antimicrobial activity was evaluated by counting the cell viability of Gram (−) and Gram (+) strains and the fungus *C. albicans*. Bacterial cells were treated with SQQ30 peptide at concentrations of 1, 5, 10, and 20 µM. The cells without the peptide were the positive control. The next day, bacterial cell survival was calculated by counting colonies [45].

The same test was carried out to evaluate the interaction between LPS and SQQ30. Bacterial cells were incubated with SQQ30 at 1 µM and with LPS at 0.5 µg/mL and co-incubated with LPS and SQQ30 at the same concentrations [46].

### 4.4. Determination of Minimal Inhibitory Concentration

The minimal inhibitory concentrations (MICs) of SQQ30 peptide and TRS21 against the strains were determined according to Heydari, M., et al., 2018 [47]; 5 × 10^5^ CFU/mL of each bacteria strain was added to 95 µL of Mueller–Hinton broth (CAM-HB; Fisher scientific, Segrate, Italy) that was supplemented or not with SQQ30 (0.1–50 µM) and TRS21 (0.5–200 µM).

The positive control was represented by gentamicin for all bacterial strains.

### 4.5. Determination of Synergism Properties

The effects of the combination of SQQ30 with TRS21 or the conventional antibiotic ciprofloxacin on *E. coli* cells were investigated using a checkerboard microdilution technique [48]. To compare SQQ30/TRS21 or SQQ30/ciprofloxacin, the drugs were applied horizontally and vertically in serial dilutions to generate combinations of two antibacterial agents at different ratios. Each test plate also contained drug-free control wells (medium only).

The *E. coli* cells, in their first log phase, were diluted, resulting in a final density of 105 CFU/mL. The FIC and fractional inhibitory concentration index (FIC I) were computed using the formula below.
$$\mathrm{FIC}\;\mathrm{of}\;\mathrm{drug}\;A=\frac{\mathrm{MIC}\;\mathrm{of}\;\mathrm{drug}\;A\;\mathrm{in}\;\mathrm{presence}\;\mathrm{of}\;\mathrm{drug}\;B}{\mathrm{MIC}\;\mathrm{of}\;\mathrm{Drug}\;A\;\left(\mathrm{alone}\right)}$$
$$\mathrm{FIC}\;\mathrm{of}\;\mathrm{drug}\;B=\frac{\mathrm{MIC}\;\mathrm{of}\;\mathrm{drug}\;B\;\mathrm{in}\;\mathrm{presence}\;\mathrm{of}\;\mathrm{drug}\;A}{\mathrm{MIC}\;\mathrm{of}\;\mathrm{Drug}\;B\;\left(\mathrm{alone}\right)}$$
FIC Index=FIC of drug A+FIC of drug B

The interaction was defined by Schelz, Z., et al., 2006, and Niu, M., et al., 2013 [48,49].

### 4.6. Fluorescence Microscopy Studies

In the dark and with shaking, 10^7^ CFU/mL of *E. coli* DH5α cells were incubated for 4 h at 37 °C with or without SQQ30 at a sub-MIC value (1 µM). Subsequently, 10 µL of the sample was treated with 4′,6-diamidino-2-phenylindole dihydrochloride (DAPI) at a concentration of 1 μg/mL and propidium iodide (PI) at a concentration of 10 μg/mL. The samples were observed as described by Di Napoli [50].

### 4.7. Scanning Electron Microscopy Studies

To observe the effects of SQQ30 on *E. coli* cells, scanning electron microscopy was used. In the presence and/or absence of SQQ30 at a sub-MIC value (1 µM), 2 × 10^8^ CFU/mL of bacterial cells were incubated for 4 h at 37 °C. After incubation, the cultures were centrifuged for 15 min at 4 °C and 7000× *g* and observed under an FEI (Hillsboro, OR, USA) Quanta 200 ESEM in high vacuum mode (P 70 Pa, HV 30 kV, WD10 mm, and spot 3.0) [51].

### 4.8. Transmission Electron Microscopy Studies

The samples were treated as previously reported [47]. At every step, the bacterial suspension (2 × 10^8^ CFU/mL) was centrifuged so as to obtain a pellet, and the supernatant was substituted with the following agent.

After a 2-day polymerization at 70 °C, resin blocks were cut into ultra-thin (50-nm-thick) sections, which were mounted on 300-mesh copper grids and observed with a Philips EM 208S TEM. TEM observations were made on *E. coli* treated with or without SQQ30 at MIC value.

### 4.9. Eukaryotic Cell Culture

HaCat (human keratinocytes) RAW 264.7 cells were cultured in Dulbecco’s Modified Eagle Medium (DMEM), supplemented with 10% fetal bovine serum, 2 mM of L-glutamine, and 1% penicillin–streptomycin. Cells were grown at 37 °C with 5% CO_2_.

### 4.10. Cytotoxicity Assays

The cytotoxic effects of SQQ30 on RAW 264.7 cells were determined by performing the MTT assay. Briefly, 2 × 10^4^ cells were seeded into a 96-well plate and incubated at 37 °C in the presence of 5% CO_2_. The medium was then replaced with 100 μL of fresh medium containing a peptide solution at a final concentration ranging from 0.15 to 20 μM/well. 

After 4 h of incubation at 37 °C, the resulting insoluble formazan salts were solubilized in 0.04 M of HCl in anhydrous isopropanol and quantified by measuring the absorbance at λ = 570 nm (SynergyTM H4, Agilent BioTek, Santa Clara, CA, USA) [52].

Cell survival was expressed as the means of the percentage values compared to the control. Analyses were performed at least 3 times.

### 4.11. Immune-Modulatory Activity of SQQ30

The ability of SQQ30 to modulate cytokines and nitric oxide production in RAW 264.7 cells was measured via ELISA (an enzyme-linked immunosorbent assay) and a Griess assay, respectively. Cells (2 × 10^4^ cells/well) were seeded into 96-well microtiter plates. The next day, the culture medium was discarded and replaced with a fresh medium either containing (i) a mixture of SQQ30 (0.1, 1, or 10 μM) and 10 ng/mL of LPS from *E. coli* 0111:B4 (co-incubation), (ii) only SQQ30 (0.1, 1, or 10 μM), or (iii) only LPS from *E. coli* 0111:B4. The LPS inhibition exerted by SQQ30 was compared to 0.1–1^−10^ μM of colistin. Cell supernatants were collected after 24 h of incubation at 37 °C and 5% CO_2_. The release of TNF-α was measured using DuoSet ELISA kits (R&D Systems, Milano, Italy), following the protocols provided by the manufacturer. 

Nitrite concentrations were determined via a colorimetric reaction using the Griess Reagent Kit for nitrite quantitation (Invitrogen™). The absorbance was measured at 548 nm using a 96-well microplate reader (SynergyTM H4, Agilent BioTek, Santa Clara, CA, USA ).

### 4.12. H_2_O_2_ Scavenging Assay

Hydrogen peroxide stability was measured following the absorbance at 240 nm of 1 mL of fresh hydrogen peroxide solution [45]. Different concentrations of SQQ30 (0–20 µM) were incubated at 20 °C in 1 mL of H_2_O_2_ solution.

The positive control was represented by resveratrol.

### 4.13. Reactive Oxygen Species (ROS) Detection Assay in HaCat Cells

HaCaT cells were seeded in a 96-well plate (1 × 10^4^ cells/well). After 24 h, the culture media were changed with fresh DMEM without phenol red supplemented with 10% FBS and with SQQ30 (1 µM and 2 µM) and/or TRS21 (25 µM and 100 µM) for 24 h. Additionally, 45 min before H_2_O_2_ treatment, 2′,7′-Dichlorofluorescin diacetate (Merk Life Science, Milano, Italy) was added to the culture media at a final concentration of 30 µM. After 45 min, the culture media were replaced with fresh DMEM without phenol red supplemented with 2% FBS.

Then, the cells were treated with 1 mM of H_2_O_2_ for 1.5 h. The fluorescence intensity (535 nm) was measured using a Synergy H4 Hybrid Microplate reader (Agilent, Santa Clara, CA, USA) [53].

## Figures and Tables

**Figure 1 antibiotics-13-00145-f001:**
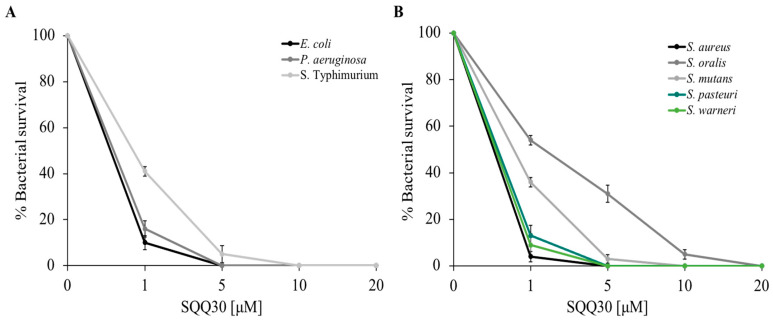
SQQ30 antimicrobial activity. Tests were performed against different bacterial strains at different concentrations (0–20 µM) in three independent experiments. Gram (−) panel (**A**) and Gram (+) panel (**B**). Standard deviations were always less than 10%.

**Figure 2 antibiotics-13-00145-f002:**
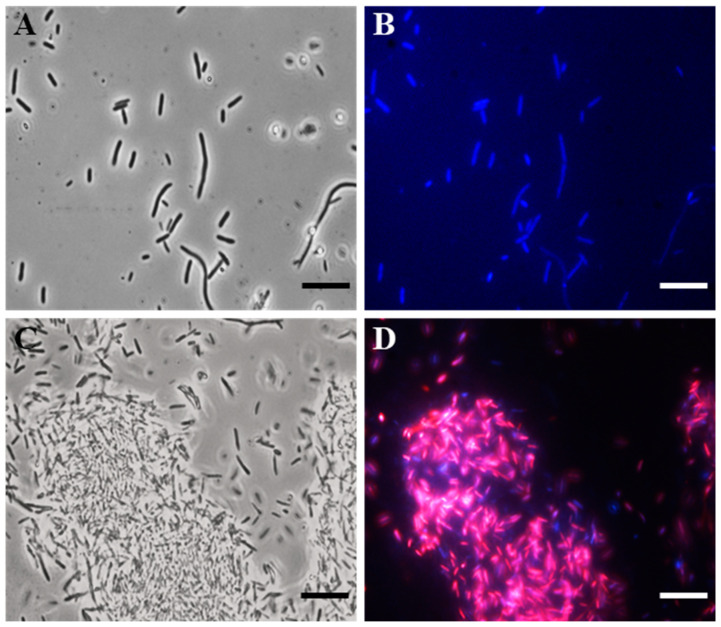
DAPI/PI dual staining images obtained via the fluorescence microscopy (FM) of *E. coli* untreated (**A**,**B**) or treated (**C**,**D**) with peptide SQQ30. Scale bars: 1 µm.

**Figure 3 antibiotics-13-00145-f003:**
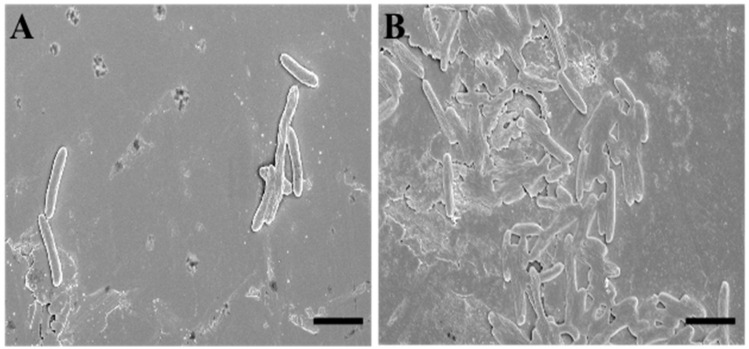
Scanning electron microscopy (SEM) images of untreated (**A**) and SQQ30-treated (**B**) *E. coli* cells. Scale bars: 5 µm.

**Figure 4 antibiotics-13-00145-f004:**
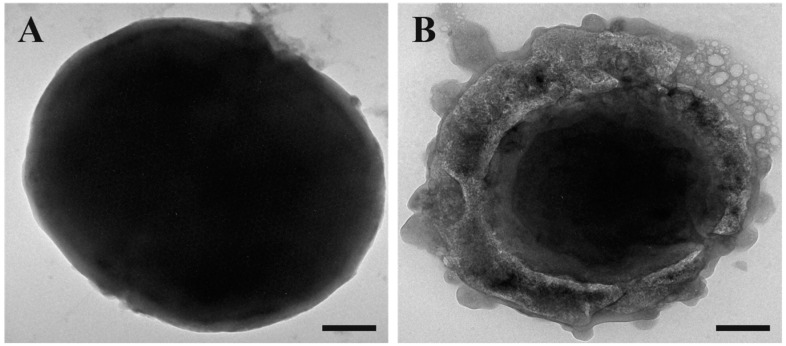
Transmission electron microscopy (TEM) images of untreated (**A**) and SQQ30-treated (**B**) *E. coli* cells. Scale bars: 200 nm.

**Figure 5 antibiotics-13-00145-f005:**
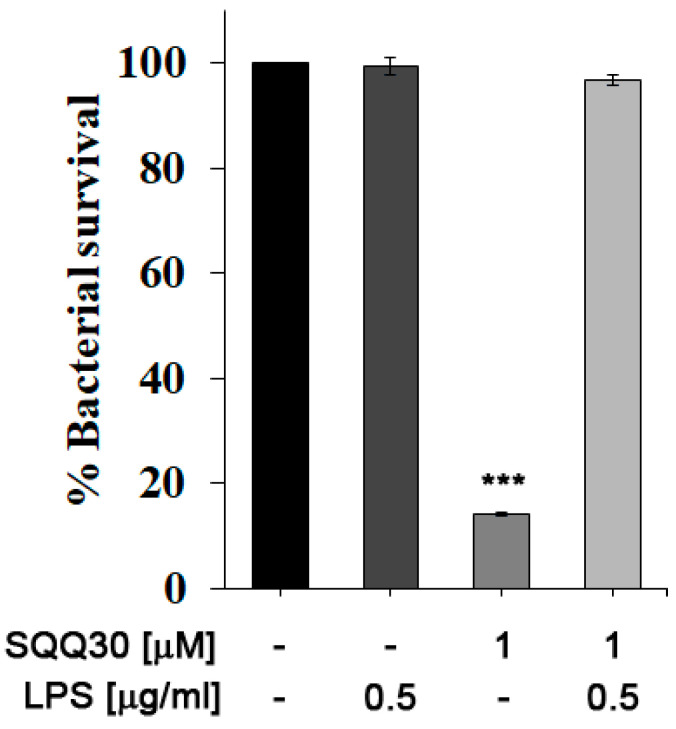
Antimicrobial assay against *E. coli*. Bacterial cells were incubated with *E. coli* LPS, SQQ30, or LPS + SQQ30. The assays were performed for three independent experiments. The error bars indicate a SE of the mean (*n* = 3). Statistical analysis was performed using a two-tailed paired *t*-test (*** *p* ≤ 0.001) versus untreated cells.

**Figure 6 antibiotics-13-00145-f006:**
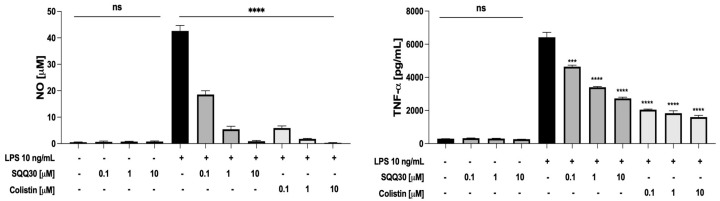
Effects of SQQ30 and colistin on the release of pro-inflammatory mediators via RAW 264.7 cells (murine macrophages). Where indicated, RAW 264.7 cells were stimulated with LPS from *E. coli* 0111:B4. The data presented were the mean values of three independent experiments. The error bars indicate ± SD values. Statistical analysis was calculated using a two-tailed paired *t*-test (ns is not significant, *** *p* < 0.001, **** *p* < 0.0001).

**Figure 7 antibiotics-13-00145-f007:**
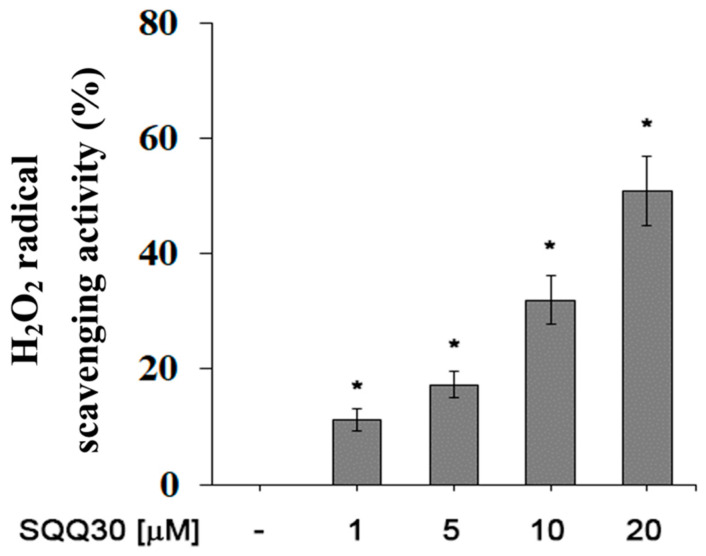
SQQ30 antioxidant activity. This graph shows the hydrogen-peroxide scavenging activity. The assays were performed for three independent experiments. The error bars indicate the SE of the mean (*n* = 3). Statistical analysis was performed using a two-tailed paired *t*-test (* *p* ≤ 0.05) versus untreated cells.

**Figure 8 antibiotics-13-00145-f008:**
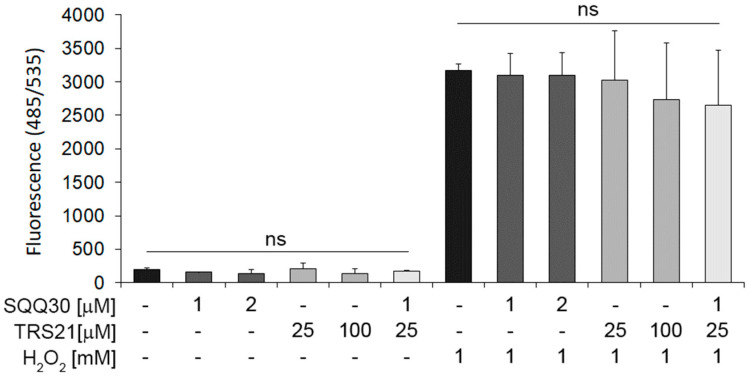
The effect of SQQ30 on oxidative stress in epithelial cells. The panel shows the reactive oxygen species (ROS) detected via fluorescence upon 24 h of treatment with the indicated peptides, with or without H_2_O_2_. The data represent the means of three experiments ± SD (ns means not significant).

**Table 1 antibiotics-13-00145-t001:** Evaluation of the minimum concentration values inhibiting the bacterial growth (MIC) of SQQ30 against model strains. The positive control was gentamicin. The values were obtained from a minimum of three independent experiments.

Strains	MIC SQQ30 [µM]	MIC Antibiotic [µM]
*E. coli*	2	23.01
*P. aeruginosa*	7	133
*S.* Typhimurium	10	61.8
*S. aureus*	1.5	14.2
*S. oralis*	18	24.8
*S. mutans*	9	16.7
*S. warneri*	6	14.64
*S. pasteuri*	7	33.48

**Table 2 antibiotics-13-00145-t002:** Determination of the FIC index. Interaction of SQQ30 with the antibiotic ciprofloxacin and the peptide TRS21 against the bacterium *E. coli*. (^a^ synergism; ^b^ additivity).

Compounds	Compounds Alone MIC [µM]	Compounds in Combination MIC [µM]	
		SQQ30 + Ciprofloxacin	SQQ30 + TRS21
SQQ30	2	1	0.5
Ciprofloxacin	1.66	0.75	-
TRS21	100	-	20
FIC index	-	0.95 ^b^	0.45 ^a^

**Table 3 antibiotics-13-00145-t003:** Concentration at 50% scavenging activity. H_2_O_2_: hydrogen peroxide. The positive control was resveratrol. The values were obtained from three independent experiments.

Sample	IC_50_ of H_2_O_2_ [µM]
SQQ30	17 ± 1
Resveratrol	0.22 ± 0.07

## Data Availability

Data are contained within the article.

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
