# Peer review of "Study of the Antimicrobial Activity of the Human Peptide SQQ30 against Pathogenic Bacteria"

_antibiotics, 2024, doi:10.3390/antibiotics13020145_

Round 1
Reviewer 1 Report
Comments and Suggestions for Authors
The authors showed the therapeutic potential of the AMP SQQ30 by using MIC assays, synergistic assays, SEM and TEM, etc.
The MIC of SQQ30 is good, but the whole quality of the manuscript is average. There are even many spelling and format problems in the current version. It is recommended to pay attention to the writing of the first draft.
First of all, the discovery or design of the peptide SQQ30 is ambiguous, as a potential antibiotic, the source needs to be clarified more clearly.
- Introduction:
Line 61: Improper use of common terms, it is recommended to change "thyroid pore" to "toroidal pore".
-2.2: Table 1: Please use the three-line table.
-2.4: The bacterial distribution density difference between the untreated and SQQ30-treated maps of Figure 2 and Figure 3 is too large. Can it provide a result map with similar bacterial density?
-2.5: line250: incorrectly marked “*** p ≤ 0.0001” instead of “*** p ≤ 0.001”
-2.6: line 280: suggested (*** p < 0,001 o **** p < 0,0001) in the middle of the" o "instead of ",".
-2.7: Figure 7, incorrectly marked “H202” instead of “H2O2”
Line 295: “as % of H2O2 removed relative to the control The assays were”, lack of punctuation marks.
-2.8: Figure 8 is illegible.
line 326: please change “The panels show” to “The panel shows”
- Discussion
line 330: Spaces in the first line are larger than four letters, please be consistent with the rest of the paragraph.
line 404: the word is misspelled, please change “promizing” to “promising”
- Materials and Methods
line 410: please change “2435,03” to “2435.03”
line 432: please get rid of “~”
line 470: please follow a comma after “To observe the effects of SQQ30 on E. coli cells”
line 486: please change “in an oven at 70 C” to “in an oven at 70 °C”
line 495 and line 513: please change “CO2” and “CO2” to “CO2”
line 536: Using scientific counting method to write “10000cells / well”
line 539: please change “H2O2” to “H2O2”
Author Response
Reviewer 1
The authors showed the therapeutic potential of the AMP SQQ30 by using MIC assays, synergistic assays, SEM and TEM, etc.
The MIC of SQQ30 is good, but the whole quality of the manuscript is average. There are even many spelling and format problems in the current version. It is recommended to pay attention to the writing of the first draft.
First of all, the discovery or design of the peptide SQQ30 is ambiguous, as a potential antibiotic, the source needs to be clarified more clearly.
Thanks for the comment. We apologize for the unclear description of the identification procedure. As the in silico method is described in two previous papers (references 17 & 20 of the previous version) and the method has already been used to identify several tens of human, animal and plant derived antimicrobial peptides, in the original version we tried to keep the length of the identification section to a minimum. In the revised version we have added more details about the method and cited a larger number of works describing the application of the method to the analysis of other representative human proteins. We hope that in the present form the identification section is more comprehensive.
- Introduction:
Line 61: Improper use of common terms, it is recommended to change "thyroid pore" to "toroidal pore".
Done
-2.2: Table 1: Please use the three-line table.
Done
-2.4: The bacterial distribution density difference between the untreated and SQQ30-treated maps of Figure 2 and Figure 3 is too large. Can it provide a result map with similar bacterial density?
Thank you for this important comment. Ten different fields of three biological replicates were counted. In general, treated cells distribution is homogenous, approximately counting 40-50 bacterial cells for each observed field. The SQQ30 treated cells bacterial distribution is not homogeneous: there are empty fields (without bacterial cells) and fields that are full of aggregated and damaged bacteria (red cells), as in figures 2 and 3. For this reason, we have shown these fields, which appear to be the most significant.
-2.5: line250: incorrectly marked “*** p ≤ 0.0001” instead of “*** p ≤ 0.001”
Done
-2.6: line 280: suggested (*** p < 0,001 o **** p < 0,0001) in the middle of the" o "instead of ",".
Done
-2.7: Figure 7, incorrectly marked “H202” instead of “H2O2”
Thanks for your comment, figure 7 has been modified and added to the text.
Line 295: “as % of H2O2 removed relative to the control The assays were”, lack of punctuation marks.
Done
-2.8: Figure 8 is illegible.
Thanks for your comment, figure 8 has been modified and added to the text.
line 326: please change “The panels show” to “The panel shows”
Done
- Discussion
line 330: Spaces in the first line are larger than four letters, please be consistent with the rest of the paragraph.
line 404: the word is misspelled, please change “promizing” to “promising”
Thanks for the comment, all changes have been made to the text.
- Materials and Methods
line 410: please change “2435,03” to “2435.03”
line 432: please get rid of “~”
line 470: please follow a comma after “To observe the effects of SQQ30 on E. coli cells”
line 486: please change “in an oven at 70 C” to “in an oven at 70 °C”
line 495 and line 513: please change “CO2” and “CO2” to “CO2”
line 536: Using scientific counting method to write “10000cells / well”
line 539: please change “H2O2” to “H2O2”
Thanks for the comment, all changes have been made to the text.
Reviewer 2 Report
Comments and Suggestions for Authors
Summary:
This manuscript delineates the discovery and characterization of a novel antimicrobial peptide (AMP) derived from the human SOGA1 protein. In light of escalating antibiotic resistance, the identification of such peptides is of significant importance. The authors have employed a bioinformatics approach to predict potential AMP regions within the human proteome and have synthesized and biologically characterized one such promising peptide, named SQQ30. The investigation encompasses assessments of antibacterial, antifungal, and immunomodulatory properties, alongside microscopic studies to elucidate the mechanism of action and evaluations of toxicity and synergistic effects with established antibiotics.
Critical Evaluations:
- Clarification of SOGA1 Identification:
The manuscript does not sufficiently detail the initial identification of the SOGA1. A more comprehensive account of the methodologies and findings leading to the discovery of SOGA1 protein, and its fragments (referenced in Figure S1) would enhance the clarity and reproducibility of the research.
2. Control Antibiotics Query:
The control antibiotics utilized in Figure 1 and Table 1 are not explicitly stated. Clarification of these controls would lend greater credibility to the analyses presented. Especially, in the context of the ensuing point down.
3. Discrepancies in MIC Values:
The reported Minimum Inhibitory Concentration (MIC) for buforin II deviates significantly from established literature values. While variations are expected due to methodological and environmental differences, such a substantial discrepancy warrants further discussion.
4. Control Peptide Selection Rationale:
The rationale behind the selection of TRS21 and ciprofloxacin as control peptides, due to their cellular, non-membrane targets, is queried. A more appropriate control, considering the origin, size, structure, and mechanism of action of SQQ30, might be LL-37. A detailed justification for the current choice or consideration of alternative controls would strengthen the study's conclusions. So, reconsider the statement in the manuscript “TRS21 and ciprofloxacin were chosen because they have the same cellular, non-membrane targets such as SQQ30”
5. LPS Binding Experiment and Conclusions:
The design and rationale of the LPS binding experiment in Section 2.5 are unclear, and the conclusions drawn appear speculative. The observed reduction in bacterial colonies following peptide treatment does not definitively indicate LPS binding. Subsequent immunomodulation experiments do suggest an interaction with LPS, yet the nature of this interaction remains ambiguous. A more rigorous experimental design and interpretation are needed.
6. Medium Composition and Peptide Activity:
The composition of the medium used for culturing cells and preparing peptide solutions, particularly concerning the presence of Fetal Bovine Serum (FBS), is not specified. As FBS can impact peptide activity, this information is crucial for interpreting the results and for future reproducibility.
7. Discussion on Synergic Effect Mechanism:
The manuscript reports the synergic effects of buforin II and SQQ30 but does not delve into potential mechanisms. An exploration of how SQQ30's membrane disruption might facilitate buforin II's cellular entry or other synergistic mechanisms would provide a more comprehensive understanding of the observed effects (can be added in the discussion section).
8. Selection of Active Peptide Region: The manuscript details the selection of a peptide region (residues 664-693) with a predicted activity value of 15.8, which is 6 residues longer than the actual signaling region (residues 665-687) with a predicted activity of 14.2. However, the manuscript lacks a detailed explanation or justification for choosing the longer peptide despite its marginally higher predicted activity. It is unclear whether the increase from 14.2 to 15.8 represents a significant enhancement in antimicrobial activity or if it justifies the increased length. A more detailed rationale for this choice, including the methodology and significance of these predicted values, would provide clarity and strengthen the choice.
9. Consideration of Higher Concentration Toxicity: The manuscript reports approximately 20% toxicity at a concentration of 20 µM for the peptide. Given that this level might be near the start of a steep increase in toxicity, it is crucial to test the peptide at higher concentrations to better understand its toxicity profile. This is especially pertinent since immunomodulatory effects were reported at 10 µM. Understanding the full toxicity curve is vital for evaluating the peptide's therapeutic potential and ensuring safety in future applications.
Author Response
Reviewer 2
Critical Evaluations:
- Clarification of SOGA1 Identification:
The manuscript does not sufficiently detail the initial identification of the SOGA1. A more comprehensive account of the methodologies and findings leading to the discovery of SOGA1 protein, and its fragments (referenced in Figure S1) would enhance the clarity and reproducibility of the research.
Thank you for the comment. As answered to reviewer 1, we apologize for the unclear description of the identification procedure. As the in silico method is described in two previous papers (references 17 & 20 of the previous version) and the method has already been used to identify several tens of human, animal and plant derived antimicrobial peptides, in the original version we tried to keep the length of the identification section to a minimum. In the revised version we have added more details about the method and cited a larger number of works describing the application of the method to the analysis of other representative human proteins. We hope that in the present form the identification section is more comprehensive.
- Control Antibiotics Query:
The control antibiotics utilized in Figure 1 and Table 1 are not explicitly stated. Clarification of these controls would lend greater credibility to the analyses presented. Especially, in the context of the ensuing point down.
Thanks for the comment. The antibiotic used in the antimicrobial activity experiments of Figure 1 and Table 1 is gentamicin. Table 1 was modified according to the journal guidelines, and the MIC value of the control antibiotic was added.
- Discrepancies in MIC Values:
The reported Minimum Inhibitory Concentration (MIC) for buforin II deviates significantly from established literature values. While variations are expected due to methodological and environmental differences, such a substantial discrepancy warrants further discussion.
Unfortunately, in the case of antimicrobial activity testing even small methodological differences might cause large differences in the results. Even in recent years comparing the data of different authors is difficult. The difficulties increase in the case of older works.
Two of the first papers on Buforins (Park et al, 1996 Biochemical And Biophysical Research Communications 218, 408–413; Park et al, 1998 Biochemical And Biophysical Research Communications 244, 253–257) report MIC values for Buforin II of 4 ug/ml corresponding to about 1.6 uM that is significantly lower than that determined by us (100 uM). In those works, the MIC values were determined by agar diffusion assay and not by the standard protocol in multiwell.
In a later paper (Park et al, 2000 Proc Natl Acad Sci U S A. 97: 8245–8250) similar values were determined by microdilution assay in 96-well plates. However, the authors used a protocol with two very relevant differences with respect to our method:
1. They used 2.5x10^4 cfu/ml (a value 20 times lower than that used by us, 5 x 10^5 cfu/ml)
2. Cells and peptides were mixed in 10 mM sodium phosphate buffer and preincubated in this buffer for 3 hours before the addition of the nutrient broth.
Both the low concentration of bacterial cells and the long preincubation at very low ionic strength can reasonably explain the lower MIC value.
In a recent paper (Lee et al. 2021 Journal of Analytical Science and Technology volume 12, Article number: 9) the MIC value on E. coli determined with a standard microdilution assay (without preincubation) was 8 uM. However, in this case the authors used LB broth instead of Mueller Hinton.
- Control Peptide Selection Rationale:
The rationale behind the selection of TRS21 and ciprofloxacin as control peptides, due to their cellular, non-membrane targets, is queried. A more appropriate control, considering the origin, size, structure, and mechanism of action of SQQ30, might be LL-37. A detailed justification for the current choice or consideration of alternative controls would strengthen the study's conclusions. So, reconsider the statement in the manuscript “TRS21 and ciprofloxacin were chosen because they have the same cellular, non-membrane targets such as SQQ30”.
We do agree with the referee; among known AMPs, LL-37 is likely a good analogue of SQQ30. However, TRS21 and ciprofloxacin were chosen in an attempt to find antimicrobials able to act synergistically with SQQ30. In general, synergy is more frequently found among molecules with different targets while drugs with the same target commonly show additive effects. We apologize for the unclear sentence. In the revised manuscript we changed it to: “TRS21 and ciprofloxacin were chosen because, differently from SQQ30 that targets membranes, they have intracellular target(s)”.
- LPS Binding Experiment and Conclusions:
The design and rationale of the LPS binding experiment in Section 2.5 are unclear, and the conclusions drawn appear speculative. The observed reduction in bacterial colonies following peptide treatment does not definitively indicate LPS binding. Subsequent immunomodulation experiments do suggest an interaction with LPS, yet the nature of this interaction remains ambiguous. A more rigorous experimental design and interpretation are needed.
Thank you for the comment. It is well known that most cationic AMPs have a significant ability to bind LPS. This is not surprising considering that they have an amphipathic structure and a high positive charge that mediate a strong binding to anionic lipids. This is also the case of LPS. Murin Raw 264.7 cells are commonly used to demonstrate the ability of antimicrobial peptides to act as LPS-scavengers, because they possess high affinity receptors for these lipids and respond to their activation by secreting several anti-inflammatory molecules. Studying the interaction in detail (e.g. determining binding constants and stoichiometry) is possible by fluorescence and/or calorimetry but, we believe, beyond the scope of the present work.
- Medium Composition and Peptide Activity:
The composition of the medium used for culturing cells and preparing peptide solutions, particularly concerning the presence of Fetal Bovine Serum (FBS), is not specified. As FBS can impact peptide activity, this information is crucial for interpreting the results and for future reproducibility.
Indeed, this information should not be overlooked. We thank the reviewer for noticing this. Medium composition and cell culture conditions were added in the methods. Regarding a possible effect of FBS on peptide activity, we tested SQQ30 on eukaryotic cells at different % of FBS (lower than 10%) and we did not observe significant effect of its activity.
- Discussion on Synergic Effect Mechanism:
The manuscript reports the synergic effects of buforin II and SQQ30 but does not delve into potential mechanisms. An exploration of how SQQ30's membrane disruption might facilitate buforin II's cellular entry or other synergistic mechanisms would provide a more comprehensive understanding of the observed effects (can be added in the discussion section).
As correctly suggested in this interesting comment, we have better clarified the effects produced by the synergism of SQQ30 with the peptide buforin II in the discussion section.
- Selection of Active Peptide Region:The manuscript details the selection of a peptide region (residues 664-693) with a predicted activity value of 15.8, which is 6 residues longer than the actual signaling region (residues 665-687) with a predicted activity of 14.2. However, the manuscript lacks a detailed explanation or justification for choosing the longer peptide despite its marginally higher predicted activity. It is unclear whether the increase from 14.2 to 15.8 represents a significant enhancement in antimicrobial activity or if it justifies the increased length. A more detailed rationale for this choice, including the methodology and significance of these predicted values, would provide clarity and strengthen the choice.
Thanks for the comment. We apologize for the unclear description. In the revised manuscript we have tried to describe more clearly the reason that led us to select the longer peptide. As argued by the reviewer scores above 13 do not necessarily indicate more active peptides. This point is more clearly described in the revised version. However, on the basis of our previous experience, even if longer peptides with higher scores are not necessarily stronger antimicrobial, usually, they are not less active. Consequently, the most prudent approach is to start from the longest sequence and make deletions guided by the in silico analysis. This approach can allow to identify the shortest peptide retaining high antimicrobial activity, an objective significant from a biotechnological point of view but, we think, outside of the scope of the present paper.
- Consideration of Higher Concentration Toxicity:The manuscript reports approximately 20% toxicity at a concentration of 20 µM for the peptide. Given that this level might be near the start of a steep increase in toxicity, it is crucial to test the peptide at higher concentrations to better understand its toxicity profile. This is especially pertinent since immunomodulatory effects were reported at 10 µM. Understanding the full toxicity curve is vital for evaluating the peptide's therapeutic potential and ensuring safety in future applications.
Thank you for the comment. We tested higher SQQ30 concentrations, respectively 50 and 100 mM. Nevertheless, we did not reach the full toxicity of the peptide.
Figure for reviewer 2. Viability of HaCat cells treated with SQQ30. Increasing concentrations of SQQ30 (from 50 μM to 100 μM) were administrated to cells for 4, 24 and 48 hours. Cell viability was determined by the MTT assay. Data represent the means of two experiments ± S.D.

Reviewer 3 Report
Comments and Suggestions for Authors
The paper is focused on the biological activity of a new AMPs identified from the sequence of the human SOGA1 protein. Some interesting properties of the SQQ30 peptide have been evidenced. However, the rationale of the paper sometimes is not easy to follow.
Some major points to be addressed are:
1) the title is focused on the antimicrobial activity of the peptide, but MBC values against the selected bacterial strains has not been evaluated;
2) why the synergistic effects have been studied for E. coli and not for human bacterial pathogens like MRSA strains?
3) why one murine and one human cell line have been used in the successive experiments? Only human cell lines should be the best choice considering the possible pharmacological applications.
4) The results of the microscopy analyses are not easy to understand. Only one magnification is showed and in the M&M section the number of bacterial cells used is not reported.
Finally, the hemolityc activity of the peptide should be investigated.
Author Response
Thanks for the comments.

Round 2
Reviewer 2 Report
Comments and Suggestions for Authors
After reviewing the authors' responses and the revised manuscript, I am satisfied with the modifications and explanations provided. However, while the authors have addressed the discrepancies in MIC values for buforin II in their response, this explanation is not reflected in the revised manuscript. I recommend that the authors include this clarification in the manuscript to prevent any potential confusion or questions from future readers. With this addition, I believe the manuscript will be suited for publication.
Reviewer 3 Report
Comments and Suggestions for Authors
The MBC has been evaluated only for E. coli (not added to the new version of the paper) and not for some of the most interesting bacterial strains, like P. aeruginosa and S. aureus.
Only one magnification has been showed for the SEM analysis, but an higher one should be added to better highlight the effects of the peptide addition. Do you have any references linked to the hypothesis that bacterial aggregation is probably due to the nature of the peptide?
The figure related to the TEM analysis is not clear. Why all the inner part of the cell is black?